# Self-Supervised Graph Representation Learning for Neuronal Morphologies

**Marissa A. Weis**[1,2,*], **Laura Hansel**[1], **Timo Lüddecke**[1], and **Alexander S. Ecker**[1,3]

[1]Institute of Computer Science and Campus Institute Data Science, University of Göttingen, Germany
[2]Institute for Theoretical Physics, University of Tübingen, Germany
[3]Max Planck Institute for Dynamics and Self-Organization, Göttingen, Germany
*_Correspondence: marissa.weis@uni-goettingen.de_

**Reviewed on OpenReview:** `https://openreview.net/forum?id=ThhMzfrd6r`

## Abstract

Unsupervised graph representation learning has recently gained interest in several application domains such as neuroscience, where modeling the diverse morphology of cell types in the brain is one of the key challenges. It is currently unknown how many excitatory cortical cell types exist and what their defining morphological features are. Here we present GRAPHDINO, a purely data-driven approach to learn low-dimensional representations of 3D neuronal morphologies from unlabeled large-scale datasets. GRAPHDINO is a novel transformer-based representation learning method for spatially-embedded graphs. To enable self-supervised learning on transformers, we (1) developed data augmentation strategies for spatially-embedded graphs, (2) adapted the positional encoding and (3) introduced a novel attention mechanism, AC-ATTENTION, which combines attention-based global interaction between nodes and classic graph convolutional processing. We show, in two different species and across multiple brain areas, that this method yields morphological cell type clusterings that are on par with manual feature-based classification by experts, but without using prior knowledge about the structural features of neurons. Moreover, it outperforms previous approaches on quantitative benchmarks predicting expert labels. Our method could potentially enable data-driven discovery of novel morphological features and cell types in large-scale datasets. It is applicable beyond neuroscience in settings where samples in a dataset are graphs and graph-level embeddings are desired.

## 1 Introduction

The brain is structured into different areas that contain diverse types of neurons (Ascoli et al., 2008). The morphology of cortical neurons is highly complex with widely varying shapes. Cell morphology has long been used to classify neurons into cell types (Ramón y Cajal, 1911), but characterizing neuronal morphologies is still a challenging open research question. Morphological analysis has traditionally been carried out by visual inspection (Ascoli et al., 2008; Defelipe et al., 2013) or by computing a set of predefined, quantitatively measurable features such as number of branching points (Uylings & Van Pelt, 2002; Scorcioni et al., 2008; Oberlaender et al., 2012; Polavaram et al., 2014; Markram et al., 2015; Lu et al., 2015; Gouwens et al., 2019). However, both approaches have deficits: expert assessments have a high variance (Defelipe et al., 2013) and the manual definition of morphological features introduces biases (Wang, 2018), thus calling for more unbiased, data-driven approaches to characterize the morphology of neurons.

Recent advances in recording technologies have greatly accelerated data collection and therefore the amount of data available (MICrONS Consortium et al., 2023; Ramaswamy et al., 2015; Scala et al., 2021; Allen Institute, 2016; Peng et al., 2021; Winnubst et al., 2019). These developments have opened the floor for data-driven approaches based on unsupervised machine learning methods (Schubert et al., 2019,

Elabbady et al., 2022). One form of data representation that is particularly suitable for neurons is representing the skeleton of a neuron as a tree. In such a tree, the root node represents the neuron's cell body and the node features are their 3D locations. The availability of a number of such skeleton datasets has recently sparked some work on graph-level representation learning of neuronal morphologies (Laturnus & Berens, 2021; Zhao et al., 2022; Chen et al., 2022a). Following this line of research and work from the graph learning community (Sun et al., 2020; You et al., 2020), we present an unsupervised graph-level representation learning approach.

Our contributions in this paper are fourfold:

1. We propose a new self-supervised model to learn graph-level embeddings for spatial graphs. Unlike previous methods, our approach does not require human annotation or manual feature definition.

2. We introduce a novel attention module that combines transformer-style attention and message passing between neighboring nodes as in graph neural networks.

3. We apply this approach to the classification of excitatory neuronal morphologies and show that it produces clusters that are comparable with known excitatory cell types obtained by manual feature-based classification and expert-labeling.

4. We outperform existing approaches based on manual feature engineering and auto-encoding in predicting expert labels.

Our code is available at `https://eckerlab.org/code/weis2023/`.

## 2 Related Work

### 2.1 Representation learning for neuronal morphologies

Morphology has been used for a long time to classify neurons by either letting experts visually inspect the cells (Ramón y Cajal, 1911; Defelipe et al., 2013) or by specifying expert-defined features that can be extracted and used as input to a classifier (Oberlaender et al., 2012; Markram et al., 2015; Kanari et al., 2017; Wang, 2018; Kanari et al., 2019; Gouwens et al., 2019) (see Armañanzas & Ascoli (2015) for review). Ascoli et al. (2008) made an effort to unify the used expert-defined features.

With the advent of new technologies for microscopic imaging, electrical recording, and molecular analysis such as Patch-seq (Cadwell et al., 2015) that allow the simultaneous recording of transcriptomy, electrophysiology and morphology of whole cells, several works have explored the prediction of cell types from multiple modalities (Gala et al., 2021) or one modality from the other (Cadwell et al., 2015; Scala et al., 2021; Gouwens et al., 2020).

Multiple previous works try to either hand-engineer or learn a representation of neuronal morphologies. Laturnus & Berens (2021) propose a generative approach involving random walks in graphs to model neuronal morphologies. Schubert et al. (2019) process 2D projections of morphologies with a convolutional neural network (CNN) to learn low-dimensional representations. Seshamani et al. (2020) extract local mesh features around spines and combine them with traditional Sholl analysis (Sholl, 1953). Gouwens et al. (2019) define a set of morphological features based on graphs and perform hierarchical clustering on them. We use the latter as a baseline for a classical approach with pre-defined features and Laturnus & Berens (2021) as a baseline of a model with learned features.

Concurrent work (Zhao et al., 2022) proposes a contrastive graph neural network to learn neuronal embeddings with a focus on retrieval efficiency from large-scale databases. Elabbady et al. (2022) learn representations of neurons based on subcellular features of the somatic region of the neurons and show that those features are sufficient for classifying cell types on large-scale EM datasets. Chen et al. (2022b) propose a combination of graph-based processing and manually-defined features to learn embeddings of neuronal morphologies using a LSTM-based network and contrastive learning. We compare to the latter in Section 5.8.

## 2.2 Graph Neural Networks (GNNs)

Graph neural networks learn node representations by recursively aggregating information from adjacent nodes as defined by the graph's structure. While early approaches date back over a decade (Scarselli et al., 2009), recently numerous new variants were introduced for (semi-) supervised settings: relying on convolution over nodes (Duvenaud et al., 2015; Hamilton et al., 2017; Kipf & Welling, 2017), using recurrence (Li et al., 2016), or making use of attention mechanisms (Veličković et al., 2018). A representation for the whole graph is often derived by a readout operation on the node representations, for instance averaging. See Dwivedi et al. (2020) for a recent benchmark on graph neural network architectures.

**Transformer-based GNNs.** Similar to us, Zhang et al. (2020) and Dwivedi & Bresson (2021) use transformer attention to work with graphs. However, Zhang et al. (2020) compute transformer attention over the nodes of sampled subgraphs, while Dwivedi & Bresson (2021) compute the attention only over local neighbors of nodes, which boils down to a weighted message passing that is conditioned on node feature similarity, and trains with supervision. Unlike these previous approaches, we compute the attention between nodes of the global graph and adapt the transformer attention to consider the adjacency matrix of the graph, which allows the model to take into account both the direct neighbors of a node as well as all other nodes in the graph. Mialon et al. (2021) consider encoding local sub-structures into their node features and leverage kernels on graphs in their attention as relative positional encodings. Their 1-step random walk (RW) kernel is similar to our AC-ATTENTION mechanism, except that the influence of the adjacency in their attention is not learnable. Ying et al. (2021) propose strategies to adapt positional encodings to graphs in order to leverage the structural information of the graphs with transformer attention. Specifically, they propose to use three different structural encodings: (1) a centrality encoding based on the node degree; (2) edge encodings based on the edge features and (3) a spatial encoding based on the shortest path between two nodes. For neural skeletons, the centrality encoding is not effective as all the nodes besides the soma have a node degree of two or three. Furthermore, the edge encoding is not applicable since in the neuronal graphs do not have edge features. We use Laplacian positional encodings instead as it was shown that they are beneficial to capture structural and positional information (Dwivedi & Bresson, 2021) and outperform previously proposed positional encodings (Zhang et al., 2020). We did not use any additional positional encodings such as shortest-path encodings (Ying et al., 2021), but they could be easily integrated into our model. Concurrent to our work, Rampášek et al. (2022) proposed a two-stream architecture, in which transformer attention and message passing are computed in parallel and then combined after each block. In contrast, we propose one combined attention mechanism that subsumes transformer attention and message passing with a learned trade-off per node between the two settings. Chen et al. (2022a) incorporate structural information into the transformer attention by extracting a subgraph representation around each node before computing attention over nodes.

**Self-supervised learning on graphs.** Self-supervised learning has proven to be a useful technique for training image feature extractors (Oord et al., 2018; Chen et al., 2020; Chen & He, 2021; Caron et al., 2021) and has been investigated for learning graph (Li et al., 2016; Hassani & Khasahmadi, 2020; Qiu et al., 2020; You et al., 2020; Xu et al., 2021) and node (Veličković et al., 2019) representations. Narayanan et al. (2017) learn graph representations through skip-gram with negative sampling by predicting present sub-graphs. You et al. (2020) propose four data augmentations for contrastive learning of graph-level embeddings. Sun et al. (2020) learn graph-level representations in a contrastive way, by predicting if a subgraph and a graph representation originate from the same graph. Similarly, Hassani & Khasahmadi (2020) put node features of one view in contrast with the graph encoding of a second view and vice versa. They build on graph diffusion networks (Klicpera et al., 2019) and only augment the structure of the graph but not the initial node features. We use Sun et al. (2020) and You et al. (2020) as a baseline for graph-level unsupervised representation learning. Qiu et al. (2020) propose a generic pre-training method which uses an InfoNCE objective (Oord et al., 2018) to learn features by telling augmented versions of one subgraph from other subgraphs with random walks as augmentations. Xu et al. (2021) aim to capture local and global structures for whole-graph representation learning. They rely on an EM-like algorithm to jointly train the assignment of graphs to hierarchical prototypes, the GNN parameters and the prototypes. Zhu et al. (2021) propose adaptive augmentation, which considers node centrality and importance to generate graph views in a contrastive

framework. Similar to our approach, Thakoor et al. (2022) use two encoders of which only one is trained and the other is an exponential moving average of the first. In contrast to our approach, though, their training objective encourages the *node embeddings* of two augmented versions of the same graph to be similar – not the *graph-level* embedding. Moreover, they use node feature and edge masking as graph augmentations.

Unlike most prior work, we contrast two global views of a graph in order to learn a whole-graph representation. Our method operates on spatially embedded graphs, in which nodes correspond to points in 3D space. We make use of this knowledge in the choice of augmentations.

## 3 GraphDINO

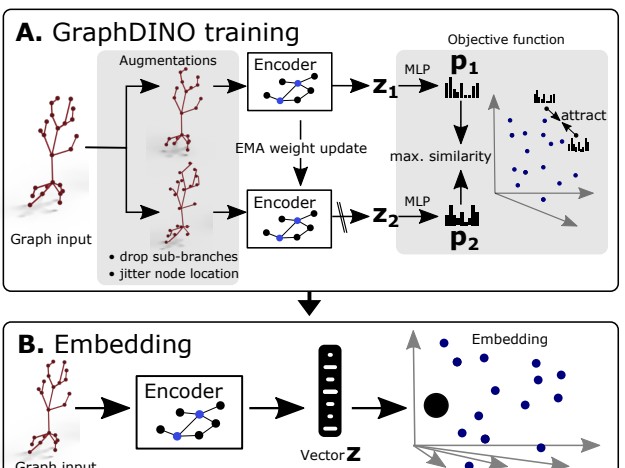

Figure 1: **A.** Self-supervised learning of low dimensional vector embeddings $z_1, z_2$ that capture the essence of the 3D morphology of individual neurons using GraphDINO. Two augmented "views" of the neuron are input into the network, where the weights of one encoder (bottom) are an exponential moving average (EMA) of the other encoder (top). The resulting latent embeddings $z$ are projected to probability distributions $p$ by a MLP. The objective is to maximize the similarity between $p_1$ and $p_2$. **B.** An individual neuron is represented by its vector embedding as a point in the $D$-dimensional vector space.

We propose GRAPHDINO, a method for self-supervised representation learning of graphs. It is inspired by recent progress in self-supervised representation learning of images that has been shown to be competitive to supervised learning without relying on labels. The core idea is to enforce that the representations of two augmented versions of the same image are close to each other in latent space.

DINO (Caron et al., 2021) is an implementation of this self-supervised learning framework consisting of two encoders with a transformer backbone. To avoid mode collapse, only one encoder is directly trained through backpropagation (student) while the weights of the other encoder (teacher) are an exponential moving average (ema) of the student's weights. The latent representations $z \in \mathbb{R}^{D_1}$ given by the encoders are mapped to probability distributions $p \in \mathbb{R}^{D_2}$ by a multi-layer perceptron (MLP) and subsequent softmax operator over which the cross-entropy loss is computed (Fig. C.4). For further explanation of DINO see Appendix C.1.

GRAPHDINO adapts this self-supervised framework to the data domain of graphs (Fig. 1). In order to use information given by the connectivity of the graph, we modify the computation of the transformer attention to take the graph adjacency matrix into account and use the graph Laplacian as positional encoding.

More specifically, we introduce the following modifications: (1) we incorporate the graph's adjacency matrix into the attention computation; (2) we use the graph Laplacian as positional encoding; (3) we define augmentations suitable for spatial graphs.

**Input.** Input to the network is the 3D shape of a neuron which is represented as an undirected graph $G = (V, E)$. $V$ is the set of nodes $\{v_i\}_{i=1}^{N}$ and $E$ the set of undirected edges $E = \{e_{ij} = (v_i, v_j)\}$ that connect two nodes $v_i, v_j$. The features of each node $v_i$ in the graph are encoded into a token using a linear transformation. These tokens are then used as input to the transformer model, which consists of $l$ multi-head attention modules with $h$ heads each.

**Attention bias.** Key-value query attention became popular in natural language modelling (Vaswani et al., 2017) and is now used routinely also in image models (Dosovitskiy et al., 2020).

To make use of the information given by the adjacency matrix $\mathbf{A} \in \mathbb{R}^{N \times N}$ of the input graph — i.e. the neighborhood of nodes —, we bias the attention towards $\mathbf{A}$ by adding a learned bias to the attention matrix that is conditioned on the input token values:

$$Attention(\mathbf{Q}, \mathbf{K}, \mathbf{V}, \mathbf{A}) = \sigma\left(\lambda \frac{\mathbf{Q}\mathbf{K}^T}{\sqrt{d_k}} + \gamma \mathbf{A}\right) \mathbf{V}, \quad \text{with } [\lambda_i, \gamma_i] = \exp(\mathbf{W}x_i), \tag{1}$$

where $\mathbf{K}$, $\mathbf{Q}$, $\mathbf{V}$ are the keys, queries and values which are computed as a learned linear projection of the tokens. $\sigma(\cdot)$ denotes the softmax function. $x_i \in \mathbb{R}^D$ is the token of node $v_i$, $\mathbf{W} \in \mathbb{R}^{2 \times D}$ is a learned weight matrix, $\lambda, \gamma \in \mathbb{R}^N$ are two factors per node that trade off how much weight is assigned to neighboring nodes versus all other nodes in the graph, and $N$ is the number of nodes.

When $\gamma = 0$ and $\lambda = 1$, the adjacency-conditioned attention (AC-ATTENTION) reduces to regular transformer attention. In the other extreme case ($\lambda = 0$,), the attention matrix is dominated by $\mathbf{A}$ and the transformer attention computation is akin to the message passing algorithm that is commonly used when working with graphs (Scarselli et al., 2009; Duvenaud et al., 2015; Gilmer et al., 2017). GRAPHDINO is more flexible than both regular message passing and point-cloud attention since it can decide how much weight is given to the neighbors of a node while maintaining the flexibility to attend to all other nodes in the graph as well.

**Positional encoding.** Following Dwivedi et al. (2020), we use the normalized graph Laplacian matrix $\mathbf{L}$ as positional encoding, which is computed by $\mathbf{L} = \mathbf{I} - \mathbf{D}^{-1/2}\mathbf{A}\mathbf{D}^{-1/2} = \mathbf{U}^T \mathbf{\Lambda} \mathbf{U}$, where $\mathbf{I}$ is the identity matrix, $\mathbf{D}$ the $N \times N$ degree matrix, $\mathbf{A}$ the adjacency matrix, and $\mathbf{U}$ and $\mathbf{\Lambda}$ are the matrices of eigenvectors and eigenvalues, respectively. The positional encodings are the first 32 eigenvectors with largest eigenvalues. Positional encodings are added to the nodes features after tokenization.

Table 1: Overview of data augmentations for spatially-embedded graphs such as neuronal skeletons.

| Level | Augmentations |
| --- | --- |
| Graph | (1) Subsampling, (2) Rotation, (5) Translation |
| |  |
| Subgraph | (3) Jittering, (4) Branch deletion |
| |  |

**Data augmentation.** Data augmentation plays an important role in self-supervised learning and needs to be adapted to the data, since it expresses which invariances should be imposed. Given the spatial neuronal data, we apply the following augmentations: **(1) Subsampling**: We subsample the original graph to a fixed number of $n$ nodes by randomly removing nodes that are not branching points (i.e. nodes connected to more than two other nodes), and connecting the two neighbors of the removed node. This facilitates batch processing. Furthermore, this augmentation retains the global structure of the neuron, while altering local structure in the two views. **(2) Rotation**: we perform random 3D rotation around the y-axis, that is orthogonal to the pia. **(3) Jittering**: we randomly translate individual node positions by adding Gaussian noise with $\mathcal{N}(0, \sigma_1)$. **(4) Subgraph deletion**: We identify branches that connect leaf nodes to the last upstream parent node in the graph, i.e. terminal branches that do not split into further branches, and randomly delete $n$ of them starting at a random location along the branch, while maintaining the overall graph structure. **(5) Graph position**: we randomly translate the graph as a whole by adding Gaussian noise with $\mathcal{N}(0, \sigma_2)$ to all nodes. Unlike Caron et al. (2021), we do not differentiate between the augmentations seen by the student and the teacher network.

## 4 Data and Experiments

### 4.1 Synthetic graphs

To demonstrate that our novel attention mechanism is strictly more powerful than simple all-to-all attention on a graph, we generate a synthetic graph dataset. In this dataset, the five classes share similar node locations but differ in how the nodes are connected. See Appendix A for the detailed generation process. We use this dataset to test the efficacy of our novel attention mechanism, AC-ATTENTION, and the positional encoding.

### 4.2 Neuronal and tree graphs

We apply GRAPHDINO to five publicly available neuronal datasets and one non-neuronal dataset.

**Blue Brain Project (BBP): Rat somatosensory cortex.** Available from the Neocortical Microcircuit Collaboration Portal of the Blue Brain Project[1] (Ramaswamy et al., 2015), the dataset contains 1,389 neurons from juvenile rat somatosensory cortex. We train GRAPHDINO without supervision on the 3D dendritic morphologies of all neurons. For evaluation, we use the subset of 616 neurons which have been labeled by experts into cell types and cortical layer. Of these 616 neurons 286 are excitatory that have been assigned to 14 cell types (Markram et al., 2015). See Appendix C.5.1 for more details on the dataset. We use this dataset to evaluate the capability of GRAPHDINO to learn useful representations of neuronal morphologies that align with known cell types, perform ablation experiments on the novel graph augmentation strategies and compare to previous work using manually-defined features.

**M1 PatchSeq: Mouse motor cortex.** The M1 PatchSeq dataset contains 275 excitatory and 371 inhibitory cells from M1 in adult mouse primary motor cortex (Scala et al., 2021).[2] The excitatory cells (M1 EXC) have been classified into tufted, untufted and other neurons based on their morphology in a previous study (Laturnus & Berens, 2021). We use this dataset to compare to previous work that learns morphological embeddings in a data-driven way. We train GRAPHDINO without supervision on the 3D dendritic morphologies of the 646 neurons. For evaluation, we follow the evaluation protocol and use the same dataset split as Laturnus & Berens (2021). We additionally report the 5-nearest neighbour accuracy of three additional dataset splits to estimate the variance due to the chosen split, since the test set is very small (60 neurons) and the balanced accuracy is strongly influenced by the morphologically heterogeneous "other" class that is only represented by six samples in the test set (Laturnus & Berens, 2021).

**Allen Brain Atlas (ACT): Mouse visual cortex.** As part of the Allen Cell Types Database, the dataset contains 510 neurons from the mouse visual cortex with a broad coverage of types, layers and transgenic lines.[3] See Allen Institute (2016) for details on how the dataset was recorded. It comes with a classification of each neuron into spiny, aspiny, or sparsely spiny, where spiny are assumed to be excitatory neurons and all else are inhibitory (Gouwens et al., 2019). Additionally, the cortical layer of each neuron is provided.

**Brain Image Library (BIL): Whole mouse brain.** The Brain Image Library contains 1,741 reconstructed neurons from cortex, claustrum, thalamus, striatum and other brain areas in mice (Peng et al., 2021).[4]

**Janelia MouseLight (JML): Whole mouse brain.** The Janelia MouseLight platform contains 1,200 projection neurons from the motor cortex, thalamus, subiculum, and hypothalamus (Winnubst et al., 2019).[5]

**Joint training on ACT, BIL and JML.** Following Chen et al. (2022b), for joint training of the ACT, BIL and JML datasets, we rotate the neurons such that the first principal component is aligned with the

---

[1]http://microcircuits.epfl.ch/#/main

[2]https://download.brainimagelibrary.org/3a/88/3a88a7687ab66069/

[3]http://celltypes.brain-map.org/

[4]https://download.brainimagelibrary.org/biccn/zeng/luo/fMOST/

[5]http://mouselight.janelia.org/

y-axis. Chen et al. (2022b) group the neurons of the three datasets ACT, BIL and JML into eleven classes based on the cortical layer or brain region they originate from. They then evaluate their learned embeddings on a subset of six (for BIL) or four classes (for ACT and JML) that have a broad coverage across the datasets. See Appendix C.5.4 for further details.

**Botanical Trees.** The Trees dataset (Seidel et al., 2021) is a highly diverse dataset comprised of 391 skeletons of trees stemming from 39 different genuses and 152 species or breedings. The skeletons were extracted from LIDAR scans of the trees. Nodes of the skeletons have a 3D coordinate associated with them. We normalize the data such that the lowest point (start of the tree trunk) is normalized to (0, 0, 0).

### 4.2.1 Data Preprocessing

Since the objective of GraphDINO is to learn purely from the 3D dendritic morphology of neurons, we normalize each graph such that the soma location is centered at (0, 0, 0) (no cortical depth information is given to the model). Furthermore, axons are removed for all experiments in the paper, because the reconstruction of axonal arbors of excitatory neurons from light microscopy images is difficult due to their small thickness and long ranges that they cover (Kanari et al., 2019) and thus often unreliable. The input nodes $V$ have features $v_i = [x, y, z]$ where $v_i \in \mathbb{R}^3$ are the spatial xyz-coordinates in micrometers [µm].

### 4.2.2 Training details.

GraphDINO is implemented in PyTorch (Paszke et al., 2019) and trained with the Adam optimizer (Kingma & Ba, 2015). The latent dimensionality of $z$ is 16 for the synthetic graphs and 32 for the neuronal and the botanical tree datasets. For M1 PatchSeq we use a latent dimensionality of 64. See C.2 for an overview of the hyperparameters used for training on the different datasets. At inference time, the latent embeddings $z$ are extracted from the student network for the unaugmented graphs. We use scipy for fitting Gaussian Mixture models (GMM) and k-nearest neighbor classifiers (kNN) (Pedregosa et al., 2011).

## 5   Results

We first establish that GraphDINO works on the synthetic graph dataset and show that our novel AC-ATTENTION is necessary for exploiting information from graph connectivity. Second, we show that our novel augmentation strategies are suitable for spatially-embedded graphs that are tree-structured and that classical GNN message-passing is not sufficient when graphs have long-ranging branches. Then, we move to the gradually more complex, biological questions of spiny-aspiny differentiation, cell type recovery and consistency with existing labels. To this end, we employ in total five neuronal datasets that encompass two species and range across multiple brain areas. Finally, we compare our model to several previous works based on manually-defined morphological features as well as approaches with learned features. See Appendix B for the application of GraphDINO to a non-neuronal dataset.

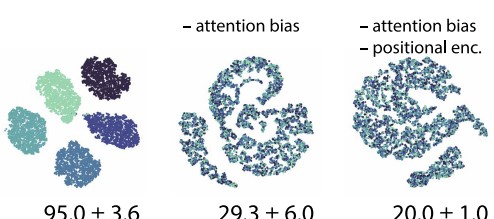

Figure 2: t-SNE embedding of latent representations of 3D synthetic graphs shown for one example run per model. Accuracy averaged over five random seeds and given as mean ± standard deviation. "–" means removing one component from the full model.

### 5.1   AC-Attention recovers information encoded by graph connectivity

We start by demonstrating the efficacy of our novel AC-ATTENTION module. For this experiment, we use the synthetic graph dataset where classes differ in how nodes are connected whereas the distribution of node positions does not vary across classes. Therefore, considering the graph structure is necessary to differentiate between the classes (more details in Appendix A). We train GraphDINO on the synthetic graph dataset without labels in three configurations: (1) with AC-ATTENTION, (2) with regular transformer attention and (3) with transformer attention and without positional encoding. We asses the quality of the learned

embedding using the ground truth labels. A linear classifier on the learned embeddings achieves a test set accuracy of (1) $95\% \pm 4$, (2) $29\% \pm 6$, and (3) $20\% \pm 1$, showing that AC-ATTENTION allows us to capture the structure of spatially embedded graphs when the location of the nodes alone does not provide sufficient information. Removing both AC-ATTENTION and the positional encoding results in the classifier performing at chance level. Using only the positional encoding performs slightly better than chance, because the positional encodings contain some information about node connections through the graph Laplacian. To make full use of the information given by the connectivity of the graphs, using AC-ATTENTION is essential (Fig. 2).

## 5.2 Tailored graph augmentations are well-suited for spatially-embedded graphs

In self-supervised learning, data augmentation is used to obtain two views that define a positive input pair. The augmentations here are chosen to encode invariances that should not change the underlying sample identity. In previous contrastive learning for graphs, these augmentations were for example dropping random edges or masking node features (You et al., 2020). These augmentations are not appropriate for our spatially-embedded graphs that form a tree and whose only node features are their 3D location in space. Thus we designed five novel augmentation techniques specifically for spatially embedded graphs such as neural morphologies or botanical trees: subsampling, rotation, node jittering, branch deletion and graph translation (see Section 3).

To test the importance of the individual graph augmentations we perform a set of ablation experiments using the BBP dataset. We remove one augmentation from our model at a time and evaluate the leave-one-out 5-nearest neighbor accuracy when predicting the expert labels. For the subsampling augmentation we vary the number of retained nodes. Our full model achieves an average accuracy of 65.8% when classifying the excitatory cells into the 12 expert

Table 2: Ablation results on the BBP dataset. Cell-type classification accuracy [%] of our model and ablations averaged over three random seeds and given as mean $\pm$ standard deviation. "–" indicates removal of an augmentation or model component.

| Model | Accuracy |
|---|---|
| **GraphDINO** | **65.8** $_{\pm 1}$ |
| – 3D rot. | 55.4 $_{\pm 1}$ |
| – node jitter | 64.8 $_{\pm 2}$ |
| – graph translation | 55.6 $_{\pm 2}$ |
| – drop branch | 64.6 $_{\pm 1}$ |
| subsampling: 50 nodes | 60.0 $_{\pm 0}$ |
| subsampling: 200 nodes | 62.0 $_{\pm 3}$ |
| – adjacency ($\gamma = 0$) | 62.8 $_{\pm 1}$ |
| – attention ($\lambda = 0$) | 59.8 $_{\pm 2}$ |

labels (Appendix C.5.1). When removing individual data augmentations the accuracy decreases (Tab. 2). Especially 3D rotation and graph translation are important augmentation strategies whose removal lead to substantial performance deterioration.

## 5.3 Message-passing is not sufficient for long-range graphs

Next, we investigate whether classical message-passing is sufficient to process graphs with long-ranging branches such as neuronal morphologies. Therefore, we train GRAPHDINO once when only using message passing while removing the global attention (setting $\lambda = 0$ in Eq. 1). This decreases the performance to 59.8% (Tab. 2). Additionally, we train INFOGRAPH (Sun et al., 2020), as a baseline for an unsupervised method that learns graph-level representations and uses GNN message-passing. INFOGRAPH achieves accuracy of 48.2% (Tab. 3). Thus, we conclude that using global at-

Table 3: Cell-type classification accuracy [%] on the BBP dataset. Performance of our model and INFOGRAPH (Sun et al., 2020) averaged over three random seeds and given as mean $\pm$ standard deviation.

| Model | Accuracy |
|---|---|
| INFOGRAPH (Sun et al., 2020) | 48.2 $_{\pm 0}$ |
| **GraphDINO** | **65.8** $_{\pm 1}$ |

tention is beneficial in situations where graphs contain long-range branches. Global attention enables information flow between distant (in terms of graph connectivity) nodes that might be close in space or function.

## 5.4 Morphological embeddings differentiate between spiny/aspiny cells and layers

To evaluate the capability of GRAPHDINO to capture essential features of 3D neuronal shapes purely data-driven, we train GRAPHDINO on the BBP dataset and use t-distributed stochastic neighbor embedding (t-SNE) (van der Maaten & Hinton, 2008) to map the learned embeddings of the BBP dataset into 2D (Fig. 3) for

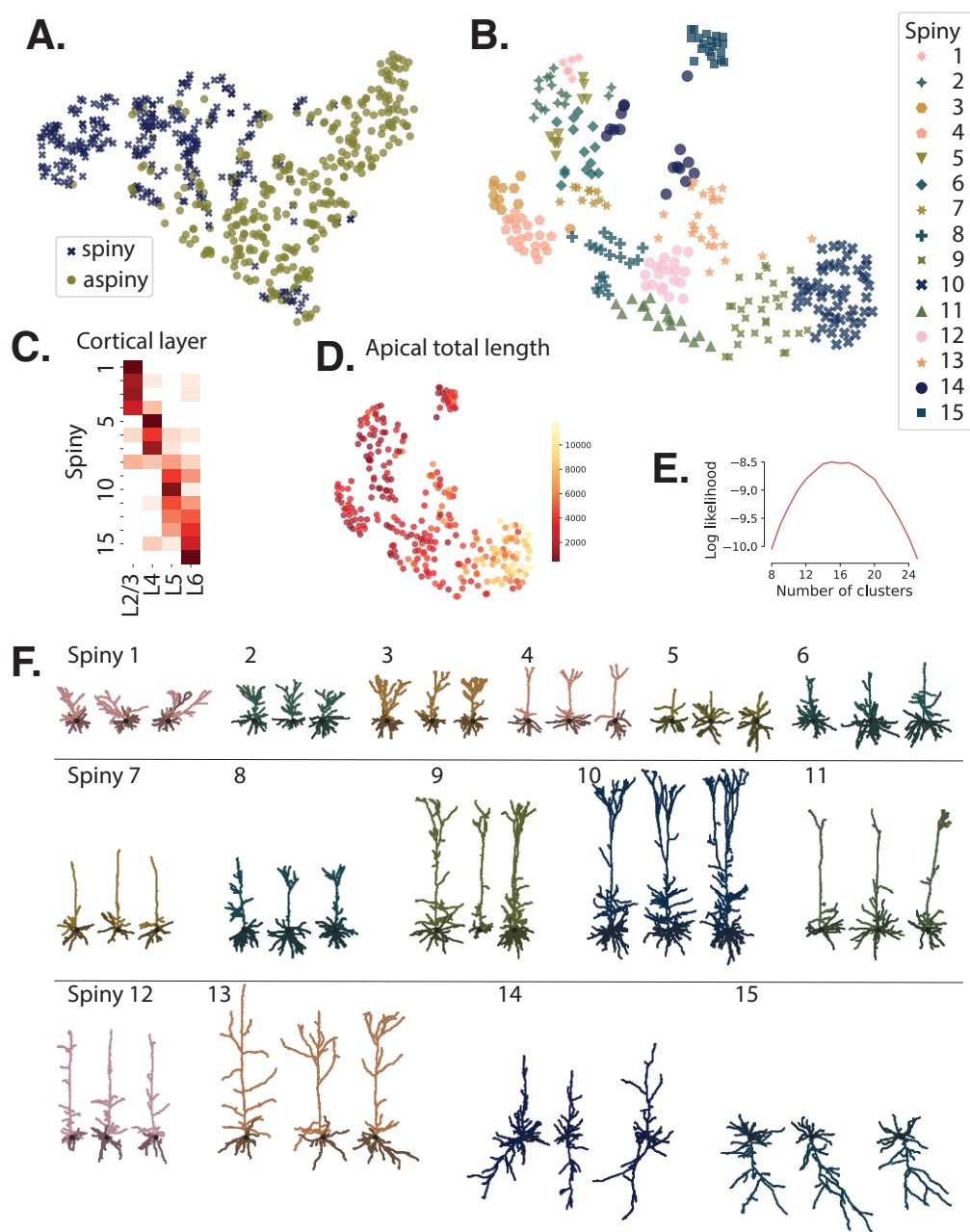

Figure 3: **A.** t-SNE embedding (perplexity=30) of latent representation of 3D neuronal morphologies of the BBP dataset showing a separation into spiny and aspiny neurons ($n = 616$). **B.** t-SNE embedding (perplexity=30) of the latent representations of the morphologies of the spiny neurons colored by the cluster found by our model ($n = 286$). **C.** Relative cortical layer distribution of neurons per cluster across L2/3—L6. Higher values are indicated by red. **D.** As **B.** but neurons colored by their total apical length revealing an organization of the latent space in terms of morphological properties. **E.** Log-likelihood of Gaussian Mixture Model on held-out test set for spiny neurons used to select the optimal number of clusters. **F.** Example neurons for each cluster are shown with apical dendrites in lighter color, while basal dendrites are colored darker. Soma is indicated by black circle.

visualization. A clear separation between spiny and aspiny neurons can be observed (see Fig. 3**A**), indicating that our learned representation captures meaningful biological differences of the neuronal morphologies.

Interestingly, some of the spiny neurons end up in the aspiny cluster (Fig. 3**A** bottom right). These are inverted L6 neurons (Fig. 3**F** Cluster 15), whose size and dendritic tree are morphologically similar to the surrounding aspiny neurons that also show a downwards bias in the dendritic tree.

### 5.5 Morphological embeddings recover known excitatory cell types

To identify cell types, we fit a Gaussian mixture model (GMM) with a diagonal covariance matrix to our learned representation of the spiny neurons. To determine the number of clusters, we fit 1,000 GMMs with different random seeds using five-fold cross-validation for 2—30 clusters. We average over the log-likelihood for each number of clusters over repetitions and folds. We find $n = 15$ to be the optimal number of clusters (Fig. 3**E**).

Having identified the optimal number of clusters, we re-fit the GMM to the full dataset including all spiny neurons. To avoid picking a particularly good or bad random clustering, we fit 100 models and choose the one that has the highest average adjusted rand index (ARI) to all other clusterings.

The spiny neurons cluster nicely into different shapes and layers (Fig. 3**F** and Appendix Fig. D.6), retrieving known excitatory cell types. The first four spiny clusters contain mainly cells from layer 2/3 (L2/3) (Fig. 3**C**) and group them by morphology: Cluster 1 contains wide and short neurons from layer 2/3, while L2/3 neurons in cluster 4 are more elongated with a less pronounced apical tuft (Fig. 3**F**). Clusters 5-7 group cells from layer 4 (L4) (Fig. 3**C**), differentiating between spiny stellate cells (cluster 5) and atufted L4 neurons (cluster 7) (Fig. 3**F**). Within layer 5 and 6, neurons are grouped by their size, amount of apical tuft and obliques, as well as the direction of the apical-like dendrites: For instance, cluster 10 groups thick-tufted pyramidal cells from layer 5 and cluster 15 contains inverted L6 neurons (Fig. 3**F**).

Most clusters show a strong preference for grouping cells whose soma position is in a certain layer (Fig. 3**C**) even though the model — in contrast to the experts who labeled the cells — does not have access to anatomical knowledge such as cortical layer of origin. One exception are pyramidal L6 cells with upward-directed apicals that separate less well and get rather clustered with L4 and L5 neurons of the same size and similar morphological shape. This is to be expected, as the model only learns to differentiate between different morphologies but has no knowledge about anatomical features such as soma depth.

### 5.6 Data-driven clusters are consistent with expert labels

To compare our data-driven features to manually-designed features, we compute the adjusted rand index (ARI) between our clusters and the expert-identified cell types on the BBP dataset and compare the performance to the clusters based on morphometrics obtained by Gouwens et al. (2020). We achieve an ARI performance of 0.31 when clustering neurons across all cortical layers together while using significantly less prior information than Gouwens et al. (2019). In comparison, Gouwens et al. (2019) reached an ARI of 0.27 with a feature space specifically designed for spiny neurons and by splitting the neurons into their cortical layer of origin before performing the clustering. This approach reduces the complexity of the problem significantly, since misassignments across layers are excluded by construction. When performing the clustering like Gouwens et al. (2019) only within the layers, we achieve an ARI of 0.46 (Tab. 4).

### 5.7 Morphological embeddings encode distinct morphological features

Laturnus & Berens (2021) classified the M1 EXC dataset (Scala et al., 2021) into three classes based on presence of an apical tuft (tufted, untufted and others). Following their work, we train a 5-nearest-neighbor classifier on our learned embeddings and show that GRAPHDINO learns meaningful features to differentiate between the three classes (Tab. 5). Our method outperforms their MORPHVAE method as well as a baseline using density maps of the neurons (Laturnus & Berens, 2021). This dataset is rather small and Laturnus & Berens (2021) used only a single train/test split. To estimate how reliable the reported accuracy metrics are, we compute the cross-validated accuracy across multiple different train/test splits, which show a vari-

Table 4: Adjusted rand index (ARI) between identified clusters and expert labels for the learned embeddings from GRAPHDINO and manually-defined features by Gouwens et al. (2019) and expert labels, when performing the clustering within cortical layers and across the whole cortex.

| Clustering | Features | ARI |
|---|---|---|
| across layers | GRAPHDINO | 0.31 |
| within layers | Gouwens et al. (2019) | 0.27 |
| | **GraphDINO** | **0.46** |

Table 5: Balanced accuracy [%] on M1 EXC test set using the learned embeddings from either GraphDINO or MorphVAE (Laturnus & Berens, 2021) (mean ± SEM across three runs and across three data splits, respectively). Percentages in brackets indicate the amount of labels used during training for MorphVAE. GraphDINO is trained without labels.

| Model | Accuracy over runs (mean ± SEM) | Accuracy over splits (mean ± SD) |
|---|---|---|
| MorphVAE (100 %) | $70_{\pm 5}$ | - |
| MorphVAE (0 %) | $58_{\pm 7}$ | - |
| Density Map (0 %) | 60 | - |
| **GraphDINO (0 %)** | $\mathbf{68}_{\pm 5}$ | $71_{\pm 9}$ |

ability across splits of ±9% (standard deviation; Tab. 5). We conclude that GraphDINO likely outperforms MorphVAE trained withoutsupervision and performs approximately on par with MorphVAE trained fully supervised.

## 5.8 Morphological embeddings encode cortical regions

TREEMOCO (Chen et al., 2022b) is an LSTM-based model that was concurrently proposed to perform unsupervised representation learning on neuronal graphs. The model uses as input the simplified skeletons of neurons that only contain the branching points as nodes. They compute 26 manually-selected features in addition to the xyz-coordinates as node features to describe the morphology of the skeletons between branching points. TREEMOCO is trained on a combination of the datasets BIL, JML and ACT and quantitatively evaluated on the task of predicting the brain anatomical region or cortical layer of origin of the neurons on a subset of the neuronal classes. Chen et al. (2022b) remove 955 neurons from the dataset due to "reconstruction errors" and evaluate on a 80-20% training-test split. Since we did not have access to the exact neurons used for training and evaluation both in terms of split and which neurons were removed, we trained unsupervised on the joint dataset and evaluated using 5-fold cross-validation, i.e. splitting the data into five folds and evaluating each fold, given the other four folds as training data and reporting the average performance across folds. For further details regarding the evaluation, see Appendix C.5.4.

GRAPHDINO performs on par or better than TREEMOCO and GRAPHCL when predicting the origin of neurons (Tab. 6). Note that GRAPHDINO is fully data-driven while TREEMOCO and GRAPHCL additionally employ manually extracted node features.

Note that the evaluation reported by Chen et al. (2022b) uses excitatory and inhibitory neurons at the same time. With this approach, morphologies of neurons of the "same" class can look very different (Fig. C.5). A better proxy task to evaluate the encoding capabilities of the models would be to restrict the evaluation to only excitatory cells. For the ACT dataset this

Table 6: Cell-type classification on the TreeMoCo dataset. Performance of our model (GRAPHDINO) averaged over three random seeds and given as mean ± standard deviation. TREEMOCO and GRAPHCL performance given as the average accuracy over the last five epochs per dataset. *Results taken Fig. C1 of Chen et al. (2022b).

| Model | BIL-6 | JML-4 | ACT | ACT spiny |
|---|---|---|---|---|
| TreeMoCo* | 76.9 | 59.7 | 53.9 | - |
| GraphCL* | 66.3 | 50.6 | 55.6 | - |
| GRAPHDINO | $79_{\pm 1}$ | $63_{\pm 6}$ | $54_{\pm 5}$ | $73_{\pm 6}$ |

information is available. We therefore repeated the evaluation only on this subset (Tab. 6), which should provide a more meaningful baseline for future studies.

## 6    Limitations

GraphDINO is designed to learn graph-level representations of spatially-embedded tree-structured graphs using self-supervised learning. As we focus on graphs where each node has a location in 3D space and design the data augmentations accordingly, the approach is not expected to work out-of-the-box on graphs that have different node features. AC-Attention is likely to be beneficial in many other scenarios as well, since it can smoothly interpolate between message passing and global attention based on node similarity, but this hypothesis remains to be tested empirically. Data augmentations would need to be adapted to the respective data domain and the respective invariances that should be encoded or supervised learning to be used. The attention mechanism is not tied in any way to the self-supervised learning objective we use.

We encode the desired invariance for neuronal morphologies in GraphDINO via tailored data augmentations. Rotation and translation equivariance could alternatively be built into the architecture of the encoders explicitly. Recent works have proposed such architectures for GNNs (Satorras et al., 2021), as well as for transformers (Fuchs et al., 2020). Adapting these for AC-Attention would be an interesting future research direction.

Computing the full transformer attention matrix has a quadratic complexity and might therefore be computationally infeasible for graphs with a large number of nodes. We solve this problem here by subsampling the neuronal skeletons to a smaller number of nodes, which has the added benefit of being a strong data augmentation that keeps the global morphology of the neuron intact while altering the local structure between the two views. However, this approach might not be suitable for all graph datasets. There has been some work in building attention mechanism that scale linearly with the number of input tokens (Wang et al., 2020; Kitaev et al., 2020; Choromanski et al., 2021), but integrating them with the message passing might not be straightforward.

Self-supervised learning has been shown to be most successful when training on large datasets (Bao et al., 2022; Oquab et al., 2023). We equipped GraphDINO with appropriate inductive biases to make it possible to learn on the smaller publicly available neuronal datasets that have been used in previous studies. Nevertheless, applying GraphDINO to neuronal datasets with more samples will likely improve its learning capabilities. With the continual development of better imaging techniques and initiatives like MICrONS (MICrONS Consortium et al., 2023) more large-scale datasets of neuronal morphologies will be available to test this hypothesis.

In terms of neuronal cell type classification, we did not take some features into account that have been previously used to differentiate cell types, such as the shape of the soma (as formerly used for GABAergic interneurons) or spine densities (Ascoli et al., 2008). Future work could focus on incorporating them into our framework. Depending on the type of feature, they could be easily integrated by adding them as features of the graph or as additional node features.

## 7    Conclusion

Increasingly large and complex datasets of neurons have given rise to the need for unbiased and quantitative approaches to cell type classification. We have demonstrated one such approach that is purely data-driven and self-supervised, and that learns a low-dimensional representation of the 3D shape of a neuron. By using self-supervised learning, we do not pre-specify which cell types to learn and which features to use, thereby reducing bias in the classification process and opening up the possibility to discover new cell types. A similar approach can also be useful in other domains beyond neuroscience, where samples of the dataset are spatial graphs and graph-level embeddings are desired, such as tree classification in forestry.

**Acknowledgments**

We thank the International Max Planck Research School for Intelligent Systems (IMPRS-IS), Tübingen, for supporting Marissa A. Weis. This project has received funding from the European Research Council (ERC) under the European Union's Horizon Europe research and innovation program (Grant agreement No. 101041669).

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

# Appendices

## A  Synthetic graph dataset

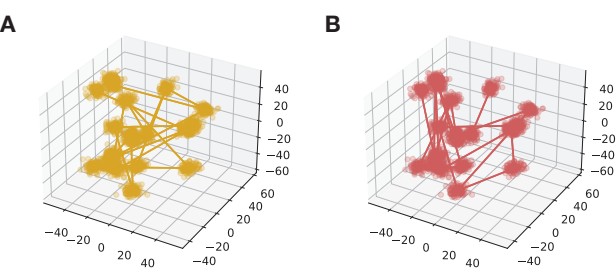

Figure A.1: Example classes 1 (A) and 2 (B) of synthetic graph dataset. Samples within one class share graph connectivity. Samples between classes share mean node locations. Node locations are drawn from $\mathcal{N}(\mu, \sigma)$.

To test whether our model is able to use information encoded in the connectivity of the graphs, we generate a synthetic graph dataset with five classes that differ in connectivity while having similar node locations. We create this synthetic graph dataset by uniformly sampling 20 mean node positions in 3D space in $[-50, 50]^3$. The mean node locations are shared between the five classes to ensure that the presence of a specific node does not encode class membership. For each class, we construct a distinct graph connectivity as follows: We first randomly sample a root node and two children, then we recursively sample one or two children per child (with a branching probability of 50%) until all 20 nodes are connected. Using this method, we generate 100,000 graphs for the training set and 10,000 graphs for validation and test set each (with $\frac{1}{5}$ class probability) by sampling node positions from $\mathcal{N}(\mu, \sigma)$ with $\mu$ equal to the above drawn means and $\sigma = 10$.

Tab. C.1, Tab. C.2 and Tab. C.3 list the hyperparamers used for experiments on the synthetic graphs.

**t-SNE of the learned latent spaces:** To visualize the learned latent space we perform t-SNE with a perplexity of 30 to reduce the embedding to two dimensions (Fig. A.2).

**Linear classifier:** We train a supervised linear classifier on the extracted embeddings of GRAPHDINO for 100 epochs and a learning rate of 0.01. To train the classifier, we use the test set that has not been used in training GRAPHDINO, and split it in 8,000 samples for training the classifier and 2,000 samples for evaluating held-out test set accuracy.

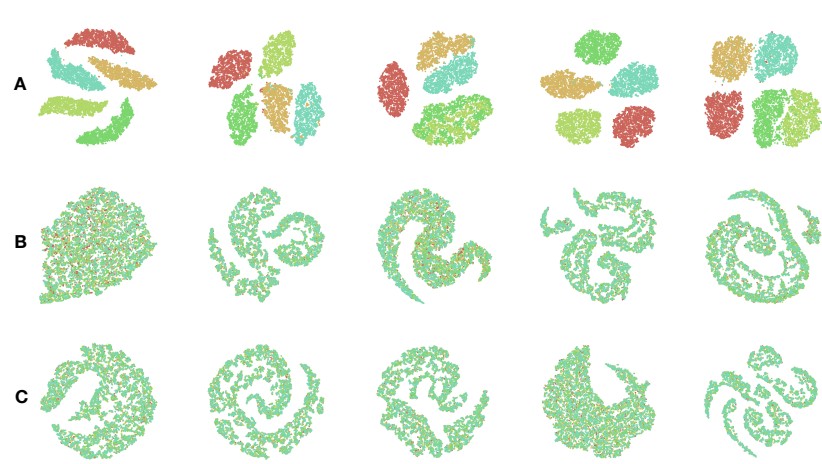

Figure A.2: t-SNE embedding of synthetic graph dataset colored by class membership for **A** five runs of GRAPHDINO with GRAPHATTENTION, **B** five runs of GRAPHDINO with regular transformer attention, and **C** five runs of GRAPHDINO with transformer attention and without positional encoding.

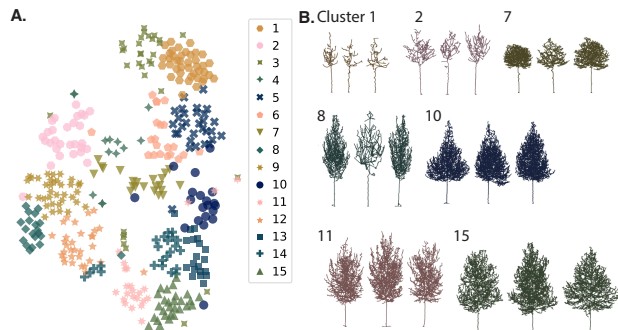

Figure B.3: **A.** t-SNE embedding of Trees dataset colored by cluster membership based on GMM clustering with 15 clusters. **B.** Three example tree morphologies are shown for different clusters.

# B  Application to different domain: Tree Morphologies

We developed a model that is able to learn graph-level embeddings of spatially-embedded graphs. So far, we have shown that it yields meaningful cell types clusterings of neuronal morphologies. To show that GRAPHDINO is applicable to data domains beyond neuronal morphologies, we train our model on 3D skeletons of individual trees (from a forest).

The Trees dataset (Seidel et al., 2021) is a highly divers dataset comprised of 391 skeletons of trees stemming from 39 different genuses and 152 species or breedings. The skeletons were extracted from LIDAR scans of the trees. Nodes of the skeletons have a 3D coordinate. We normalize the data such that the lowest point (start of the tree trunk) is normalized to (0, 0, 0).

GRAPHDINO learns a latent space that orders tree morphologies with respect to their size, crown size and crown shape (Fig. B.3, Fig. D.7).

# C  Extended Methods

## C.1  Background: DINO

DINO (Fig. C.4) (Caron et al., 2021) is a method for self-supervised image representation learning. Similar to previous approaches, it consists of two image encoders which process different views of an image. These views are obtained by image augmentation. The training objective is to enforce both encoders to generate the same output distribution when the same input image is shown. This can be implemented by the cross entropy loss function: $\sum_i -q_i \log p_i$. Both encoders are transformers that share the architecture but differ in their weights: One of the encoders is the student encoder which receives weight updates through gradients of the training objective while the other encoder's (teacher) weights are an exponential moving average of the student's weights. In contrast to some other self-supervised methods, DINO does not require contrastive (negative) samples. To prevent collapse, i.e. predicting the same distribution independent of the input image, two additional operations on the teacher's predictions are crucial: sharpening by adjusting the softmax temperature, and centering using batch statistics. Besides competitive performance on downstream image classification tasks, another key finding of the paper is that object segmentations emerge in the self-attention when applying DINO training on visual transformer image encoders.

## C.2  Data preprocessing.

To speed up data loading during training, we reduce the number of nodes in the graph of each neuron to 1000 nodes in the same way as when subsampling and ensure that it contains only one connected component. If there are unconnected components, we connect them by adding an edge between two nodes of two unconnected components that have the least distance between their spatial coordinates.

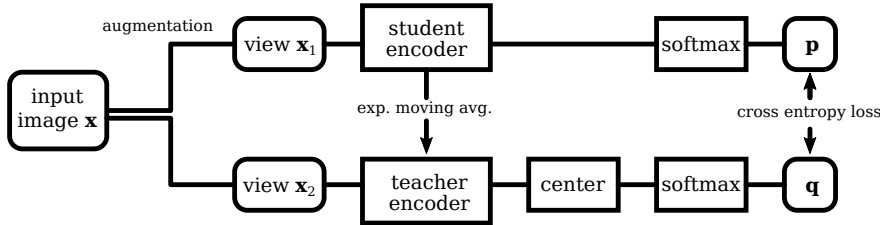

Figure C.4: The DINO method for self-supervised image representation learning (figure adapted from Caron et al. (2021)).

### C.3 Training details and hyperparameters

To select hyperparameters we run three grid searches and pick the best hyperparameters according to the lowest average loss over the BBP and M1 PatchSeq dataset.

For the optimization, we run a hyperparameter search over batch size $\in \{32, 64, 128\}$, learning rate $\in \{10^{-3}, 10^{-4}, 10^{-5}\}$, and number of training iterations $\in \{20,000, 50,000, 100,000\}$.

For the augmentation strength, we run a hyperparameter search over jitter variance $\sigma_1 \in \{1.0, 0.1, 0.001\}$, number of deleted branches $n \in \{1, 5, 10\}$, and graph position variance $\sigma_2 \in \{0.1, 1.0, 10.0\}$.

For the architecture, we run a hyperparameter search over latent dimension $\in \{16, 32, 64\}$, number of GRAPHATTENTION blocks (depth) $\in \{5, 7, 10\}$, and number of attention heads per block $\in \{2, 4, 8\}$.

#### C.3.1 Architecture Hyperparameters

Table C.1: Hyperparameters used for the architecture for the different datasets. $\mathbf{D_1}$: Dimensionality of latent embedding $z$. $\mathbf{D_2}$: Dimensionality of probability distribution $p$. **PE:** Positional encoding. **T temp:** Softmax temperature of teacher network.

| Dataset | $\mathbf{D_1}$ | $\mathbf{D_2}$ | # layers | # heads | MLP dims | PE dims | T temp |
|---|---|---|---|---|---|---|---|
| Synthetic Graphs | 16 | 300 | 4 | 4 | 16 | 16 | 0.04 |
| BBP | 32 | 1000 | 10 | 8 | 64 | 32 | 0.06 |
| M1 PatchSeq | 64 | 1000 | 7 | 8 | 64 | 32 | 0.06 |
| Joint dataset (BIL, JML, ACT) | 32 | 1000 | 7 | 4 | 64 | 32 | 0.06 |
| Trees | 32 | 1000 | 7 | 8 | 64 | 32 | 0.06 |

Tab. C.1 lists the hyperparameters used for the architecture for the different datasets. For the synthetic graph dataset, we downscale the network as it is a simpler dataset. DINO (Caron et al., 2021) uses an output dimensionality of 65,536 for $p$ when training on ImageNet (Deng et al., 2009) (1,000 classes). The number of classes in the neuronal datasets is unknown, but previous literature described $14-19$ cell types (Gouwens et al., 2019; Markram et al., 2015). Hence, we decrease the number of dimensions $D_2$ of $p$ proportionally to 1,000, approximately retaining the ratio between classes and number of dimensions.

#### C.3.2 Optimization Hyperparameters

The learning rate is linearly increased to the value given in Tab. C.2 during the first 2,000 iterations and then decayed using a exponential decay with rate 0.5 (Loshchilov & Hutter, 2016).

Table C.2: Hyperparameters used for optimization for the different datasets.

| Dataset | Iterations | Batch size | Learning rate |
|---|---|---|---|
| Synthetic Graphs | 100,000 | 512 | $10^{-4}$ |
| BBP | 100,000 | 64 | $10^{-4}$ |
| M1 PatchSeq | 50,000 | 128 | $10^{-3}$ |
| Joint dataset (BIL, JML, ACT) | 100,000 | 128 | $10^{-3}$ |
| Trees | 100,000 | 64 | $10^{-4}$ |

### C.3.3 Augmentation Hyperparameters

Table C.3: Augmentation hyperparameters for the different datasets. **# nodes:** Number of nodes to subsample to. $\sigma_1$: Variance of node jittering. **# DB:** Number of deleted branches. $\sigma_2$: Variance of graph translation.

| Dataset | # nodes | $\sigma_1$ | # DB | $\sigma_2$ |
|---|---|---|---|---|
| Synthetic Graphs | 15 | 0.1 | 0 | 0 |
| BBP | 100 | 0.001 | 10 | 10.0 |
| M1 PatchSeq | 100 | 0.1 | 10 | 10.0 |
| Joint dataset (BIL, JML, ACT) | 200 | 1.0 | 5 | 10.0 |
| Trees | 200 | 0.1 | 5 | 10.0 |

### C.3.4 Computation

All trainings were performed on a NVIDIA Quadro RTX 5000 single GPU. Training on the neuronal BBP dataset ran for approximately 10 hours for 100,000 training iterations.

### C.4 Inference

To extract the latent representation per sample, we encode the unaugmented graphs subsampled to 200 nodes using the student encoder and extract the latent representation $z$ using the weights of the last iteration of training (no early-stopping is used).

### C.5 Evaluation

### C.5.1 Evaluation on BBP

For visualization of the latent space, we use t-distributed stochastic neighbor embedding (t-SNE) (van der Maaten & Hinton, 2008) with PCA-initialization, Euclidean distance and a perplexity of 30.

For quantitative evaluation we use the subset of labeled excitatory neurons ($n = 286$) with the following 14 expert labels: L23-PC, L4-PC, L4-SP, L4-SS, L5-STPC, L5-TTPC1, L5-TTPC2, L5-UTPC, L6-BPC, L6-IPC, L6-TPC-L1, L6-TPC-L4, L6-UTPC, L6-HPC (Markram et al., 2015).

For the ablation experiments and the comparison to INFOGRAPH Sun et al. (2020), we perform k-nearest neighbor (kNN) classification with $k = 5$ in a leave-one-out setting predicting the above listed expert labels with two exceptions: We remoev the L6-HPC cells, since there are only three samples in the dataset, and we group the L5-TTPC1 and L5-TTPC2 into one class L5-TTPC following previous work that found that they rather form a continuum then two separate classes (Gouwens et al., 2019; Kanari et al., 2019).

For the clustering analysis and the comparison to Gouwens et al. (2019), we follow Gouwens et al. (2019) and compute the adjusted rand index between our found clusters and the 14 expert labels. To determine the optimal number of clusters, we use cross-validation to compute the log-likelihood of held-out data of the Gaussian Mixture model and choose the number of clusters with the highest log-likelihood. The optimal

number of clusters is 15 for the BBP dataset. To perform clustering within cortical layers, we chose the number of clusters per layer based on the number of clusters with the majority of cells from the cortex-wide clustering (Fig. 3): four for layer 2/3, layer 5 and layer 6 and three for layer 4.

### C.5.2 Comparison to InfoGraph (Sun et al., 2020)

We use the official implementation[6] to train INFOGRAPH on the BBP dataset. We perform a hyperparameter search for INFOGRAPH as detailed in the original publication (Sun et al., 2020) and extend it to include more training epochs to train it for approximately the same number of iterations as GRAPHDINO. In detail, we run a grid search over learning rate (lr) $\in \{10^{-2}, 10^{-3}, 10^{-4}\}$, number of training epochs $\in \{10, 20, 100, 200, 1,000, 2,000\}$ and GNN layers $\in \{4, 8, 12\}$. We select the hyperparameters based on the lowest unsupervised loss. The chosen hyperparameters are: $lr = 0.001$, $epochs = 1,000$ and four GNN layers with a hidden dimensionality of 32.

We evaluate the performance of INFOGRAPH (Sun et al., 2020) using a kNN classifier analogous to the ablation experiments (Appendix. C.5.1).

### C.5.3 Comparison to MorphVAE (Laturnus & Berens, 2021)

We follow the evaluation protocol of Laturnus & Berens (2021) and perform k-nearest neighbor (kNN) classification with $k = 5$ on the learned latent embeddings of the excitatory neurons to predict whether they are untufted, tufted or "other" on the test set ($n = 60$) and report the balanced accuracy. The "other" class only contains six examples in the test set. To get an estimate of the variance that is due to the chosen data split, we additionally evaluate three further data splits and report the average test set performance over the three splits. We report the performance of MORPHVAE as given in Tab. 3 of Laturnus & Berens (2021).

### C.5.4 Comparison to TreeMoCo (Chen et al., 2022b)

A fair comparison to TreeMoCo proved difficult. We tried to replicate their setting as best as possible from the information given in the paper as well as by inferring it from their code base[7] while trying to set up a more fair benchmark for future works.

We downloaded the three datasets BIL, JML and ATC using the official code base of TREEMOCO and used it to assign the eleven class labels: L1, L2/3, L4, L5, L6, VPM, CP, VPL, SUB, PRE, MG and Others as used by Chen et al. (2022b). However, our cell counts slightly differ from those given in Chen et al. (2022b). More specifically, the JML dataset contained 1,200 neurons instead of 1,107.

Chen et al. (2022b) removed a substantial amount of the neurons (995 of 3,358 neurons) from the datasets due to reconstruction errors. Since we did not have access to the identities of these neurons, we trained GRAPHDINO unsupervised on all cells with more than 200 nodes ($n_{total} = 3,138$; $n_{BIL} = 1,739$, $n_{JML} = 889$, $n_{ACT} = 510$) and evaluated the proposed classes as assigned by the TREEMOCO code base. We replicated the proposed data preprocessing by centering the somata at $(0, 0, 0)$ and aligning the neurons' first principal component to the y-axis.

Chen et al. (2022b) performs the quantitative evaluation on a 80-20% training-test data split. Since we did not have access to the exact split, we performed five cross-validations instead and report the average accuracy over folds.

According to the paper, Chen et al. (2022b) perform k-nearest neighbor classification ($k = 5$ or $k = 20$ depending on the dataset). We unify the evaluation and report the kNN accuracy with $k = 5$ for all experiments in this paper. For reference, we list the $k = 20$ performance in Tab. C.4. In their code base, the implementation of kNN is weighted, where the neighbors vote is weighted by the cosine similarity of the embeddings. We follow the description in the paper (Chen et al., 2022b) and use the standard kNN classification without weighing the neighbors' votes.

---

[6] https://github.com/sunfanyunn/InfoGraph
[7] We additionally tried to reach out to the authors but did not get a reply.

Table C.4: Cell-type classification on the TreeMoCo dataset. Performance of our model (GRAPHDINO) averaged over three random seeds and given as mean ± standard deviation when using $k = 20$ for the kNN classifer.

| Model | BIL-6 | ACT |
|-------|-------|-----|
| Ours | 78 ± 2 | 54 ± 4 |

Figure C.5: Example neurons labeled as Isocortex 4 of the ACT dataset.

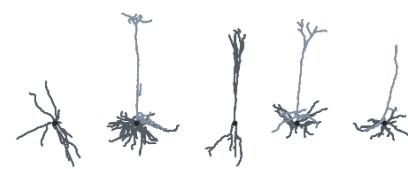

The performances reported by Chen et al. (2022b) are overfitted on the test set: They picked the best test set performance over epochs (for the three datasets separately) (see Fig. C1 in Chen et al. (2022b)). Additionally, they picked whether to use the latent embedding $z$ or the projection head's output $p$ based on the test set performance per dataset. To give an estimate of the less overfitted performance of TREEMOCO (Chen et al., 2022b) (at least with respect to which epoch to evaluate), we report the averaged performance over the last five epochs given by Fig. C1 (Chen et al., 2022b).

Similarly, the performance of GRAPHCL (You et al., 2020) as reported by Chen et al. (2022b) is picked as the best test set performance per dataset over training epochs. We therefore report the average accuracy over the last five epochs with the given by Fig. C1 (Chen et al., 2022b).

## D   Complete cluster visualizations

In the Fig. D.6 and Fig. D.7, we show the cluster assignments of all samples of the excitatory BBP dataset ($n = 286$) and the Trees dataset ($n = 391$), respectively.

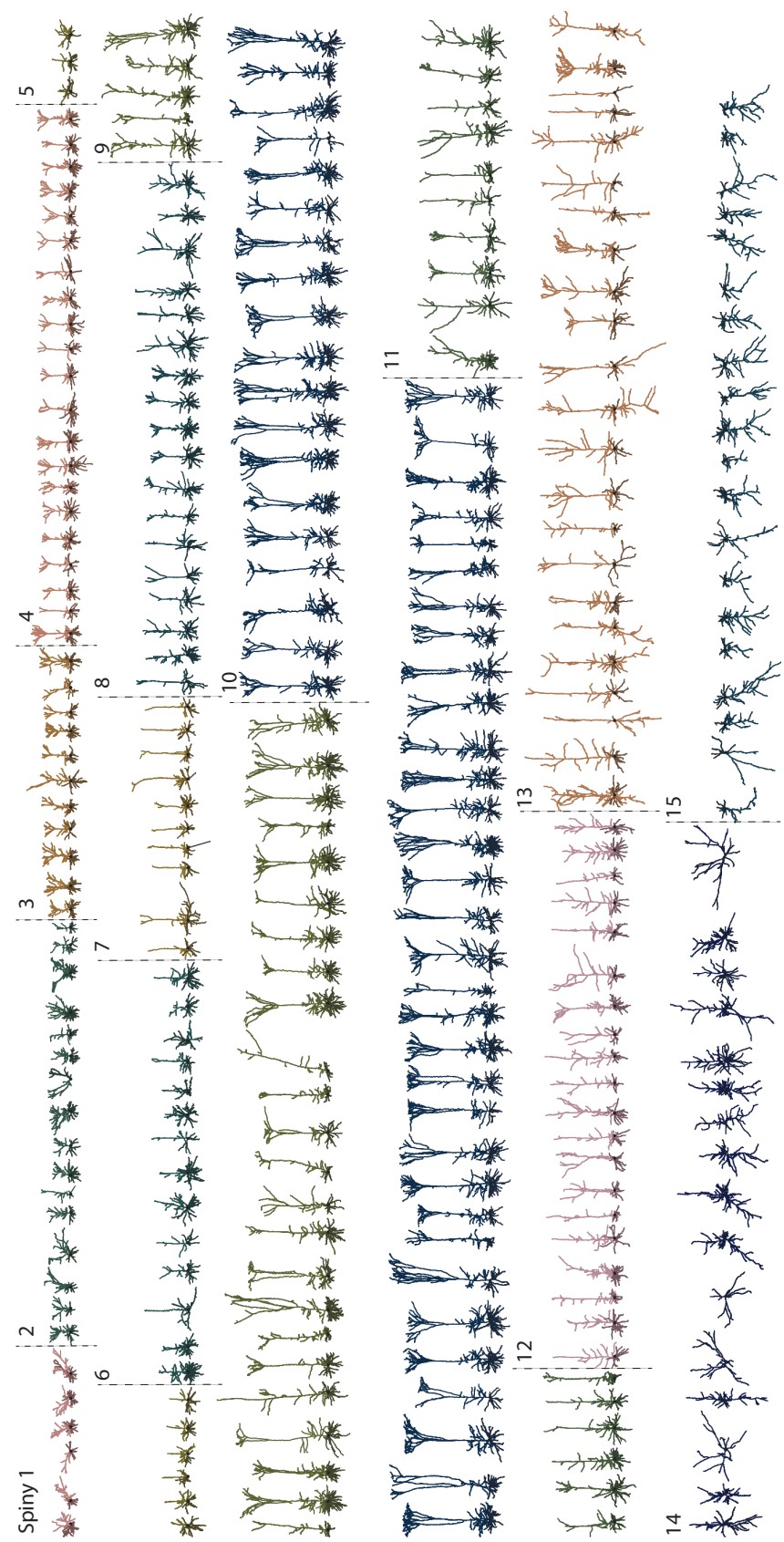

Figure D.6: Clusters of spiny neurons of BBP dataset as identified by GMM based on our learned feature space. Apical dendrites are colored lighter, while basal dendrites are shown in a darker color. Soma is marked by a black circle.

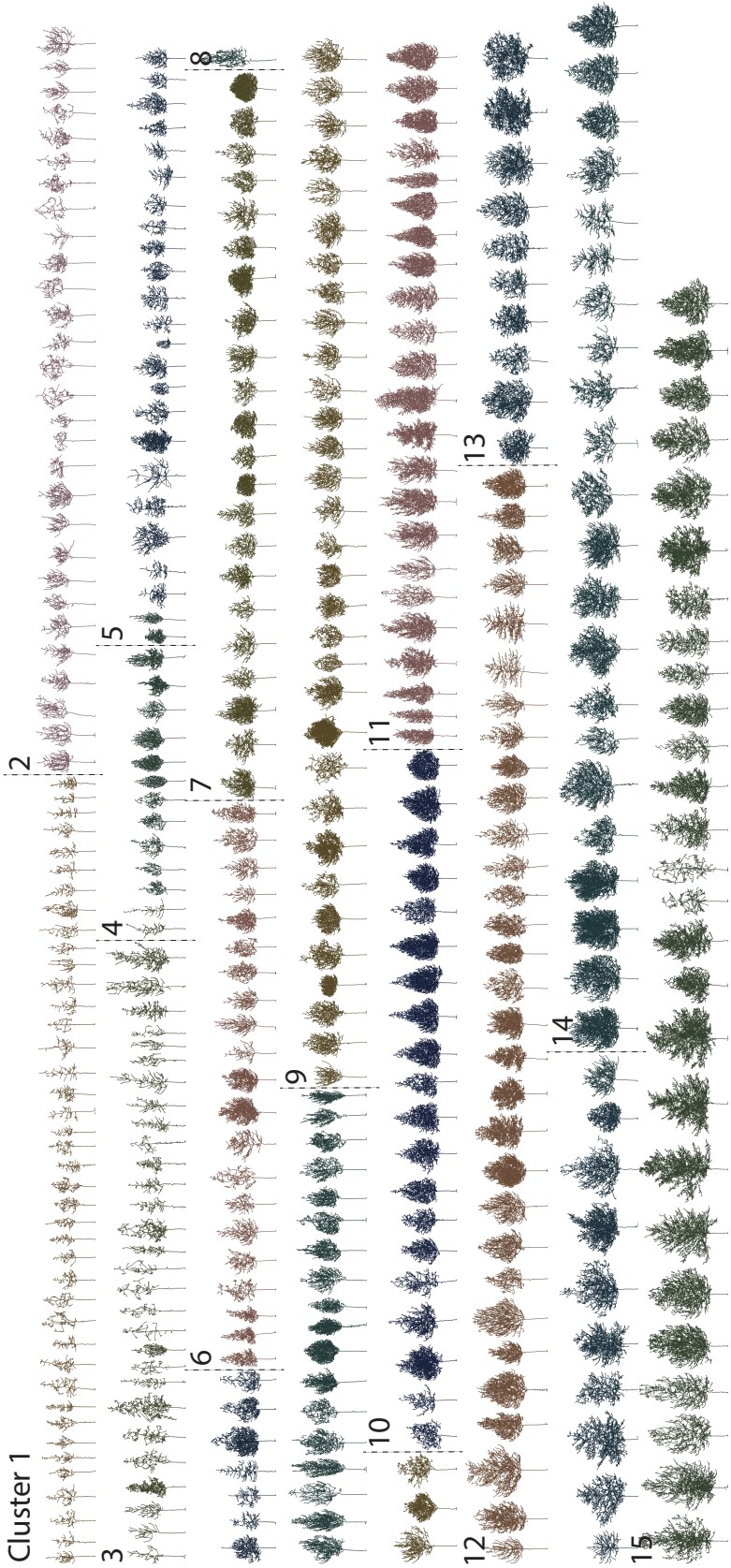

Figure D.7: Clusters of trees as identified by GMM based on our learned feature space.

