# OpenReview forum: "Self-Supervised Graph Representation Learning for Neuronal Morphologies"
_TMLR — Accepted by TMLR_

### Review · Reviewer_ooUU · 2023-04-02

**Summary Of Contributions:**

This paper introduces a new self-supervised method for graph representation learning. This learning method is specifically designed for neuronal morphologies representation learning, but can be generalized to other similar graphs to some extent. The contributions of this paper mainly include: (1) a new self-supervised graph-level feature learning method; (2) a new attention module is introduced; (3) this paper offers a bunch of ablation studies and visualizations, which demonstrate the effectiveness of the proposed method. Overall, the findings of this paper are interesting.

**Audience:**

Yes

**Broader Impact Concerns:**

I do not see any ethical issues of this work.

**Claims And Evidence:**

Yes

**Requested Changes:**

One interesting question is that: can we pre-train the model on the synthetic dataset and then fine-tune it on another small dataset? Is it possible that this pre-training will further boost the performance?

**Strengths And Weaknesses:**

Strength:
1. This paper is well written, with clear motivations and organizations.
2. The proposed method is novel to some extent, with tailored data augmentation methods for the neuronal skeleton graph (and other similar graphs).
3. This paper offers a lot of ablation studies and visualizations. They are helpful in understanding the benefits of the proposed method, and demonstrate the advantages of this method compared to the prior works.

Weakness:
1. The novelty of the proposed method is not very significant. The overall idea is built upon DINO. The positional encoding strategy is borrowed from previous work. The major novelty lies in the data augmentation, as well as in the hybrid combination of attention and adjacency matrix. Overall I think such technical novelties are not significant.
2. Usually for the DINO method on image domain, it needs a lot of training data. In this paper I find the proposed method was only trained on a few hundred of graphs (e.g., 646). It is surprising that the trained model on such a small dataset still works quite well. Is this due to some kind of overfitting? Or is it possible that the task is not challenging enough? Can we pre-train the model on the synthetic dataset and then fine-tune it on this small dataset (in a self-supervised manner)?

---

> ### Author Response · Authors · 2023-05-31
> **Authors’ response to Reviewer ooUU**
>
> Thank you for your feedback and assessing our work as interesting.
>
> **Dataset size**
>
> We agree that transformer architectures and SSL for computer vision work best in a setting with large datasets, as shown exemplary by DINO. For transformers, that’s not unexpected, because they do not share the strong inductive bias of convnets and, hence, have to learn the spatial structure of images from scratch. For SSL method, keep in mind that the strong supervised baselines have also been trained on large datasets, and more data almost always helps. It is not clear a priori why SSL would *require* large datasets to work. Model architectures with appropriate inductive biases for the image domain, such as convolutional neural networks, can be trained much more data efficiently. Similarly, we introduce adjacency-conditioned attention as well as suitable data augmentations to instill our model with the appropriate inductive biases for our data domain, which makes smaller datasets less of a concern for us, compared to DINO on images.
>
> To make sure that our reported results are not overfitted, we report the quantitative performance of our model using either cross-validation or held-out test sets (depending on the setting of previous works to make it comparable).
> Additionally, we show that GraphDINO performs well on the synthetic dataset when trained with more data and evaluated on held-out test data.
>
> **Pretraining on the synthetic data**
>
> Pretraining on the synthetic data is unfortunately not possible. Currently no good generative model of neuronal morphologies exists. This is an ongoing research direction.
> Our synthetic dataset was specifically generated to assess the capabilities of GraphDINO to extract information from graph connectivity, but does not follow neuronal skeletons statistics. Thus, it is not suitable data for pretraining.

---

### Review · Reviewer_wWM4 · 2023-04-21

**Summary Of Contributions:**

This work proposes a representation method for 3D neuronal morphologies. Seeing neuron skeletons as spatial trees, the paper proposes (i) modification to the transformer architecture suited to graph processing (ii) domain specific data augmentations to be used in the context of DINO (iii) extensive study of the resulting representation.

(i) Self-attention is modified so that it interpolates between global attention and attention limited to neighboring nodes only. This interpolation is learned and depends on the node features.

(ii) Data augmentations are central in non-contrastive SSL since these methods enforce similar embeddings for two different augmented versions of the same sample. Therefore, data augmentation determines to some extent to what modification the final representation will be invariant. Domain specific data augmentations are therefore crucial. Here, the authors argue that their embedding should be invariant to (a) tree specific subsampling (b) 3D rotation (c) Gaussian noise added to node coordinates (d) tree specific subgraph deletion (e) random translations of the node coordinates.

(iii) The authors demonstrate that the learned representation retains many characteristics of interest for practitioners, e.g., recovering known excitatory cell types, or forming clusters that are consistent with expert labels. The learned representations seem to outperform existing baselines. They also perform ablations on their proposed augmentation and attention strategies.


**Audience:**

Yes

**Claims And Evidence:**

Yes

**Requested Changes:**

See above (all equally important):
- Discussing some missing work, or cited work that haven't been used here.
- Discussing limitations and potential improvements.
- Clarification of node feature masking.

**Strengths And Weaknesses:**

Strengths:

- This work makes good use of the existing knowledge on SSL and graph transformers:  proposed strategies for domain specific data augmentation and attention on neuronal graphs make sense.
- The authors did a good job at demonstrating the usefulness of their representation. SSL and graph transformers seem a promising direction for studying neuronal morphologies and perhaps making discoveries from large, unlabelled datasets.
- The paper is overall well written.

Weaknesses:
- The authors could have done a slightly better job at discussing the relevant literature. For example:
  - In SSL for graphs, [1] adapted BYOL to graph representation learning.
  - The authors mention various works proposing graph transformers. These works contain schemes that interpolate between local and global attention, as well as different position encodingss since Laplacian eigenvectors suffer from issues: why are they not suited to the problem of learning representation of neuronal morphologies?


- In terms of employed methods, there seems to be some room for improvement that could also be discussed in the limitations.
  - Taken separately, transformers and SSL are often data hungry. Here, the data only consists of a few thousands of examples: although the need for data is alleviated here since the authors inject some inductive biases in the architecture (namely the adjacency matrix in attention and a non-learned positional encoding), have you considered looking for more data? This may lead to substantial improvements. This could also improve the instability of the results (error bars seem a bit high).
  - In computer vision, having a balanced hence curated dataset (in terms of underlying class) is crucial to achieve state of the art performance with SSL. Have you considered this issue here?
  - The authors seek a representation that is invariant to translations and 3D rotations. Instead of enforcing this property through SSL data augmentation, have you considered using equivariant architectures such as [2][3]? ([2] can be adapted to transformers). This added inductive bias could also alleviate the need for data.

- Could you clarify why masking node features was not a good augmentation strategy in your setup? In SSL, masking is generally simple and beneficial.

[1] Thakoor et al. Large-Scale Representation Learning on Graphs via Bootstrapping.

[2] Victor Garcia Satorras, Emiel Hoogeboom, and Max Welling. E (n) equivariant graph neural networks.

[3] Fabian Fuchs, Daniel Worrall, Volker Fischer, and Max Welling. Se (3)-transformers: 3d roto-translation equivariant attention networks.

---

> ### Author Response · Authors · 2023-05-31
> **Authors’ response to Reviewer wWM4**
>
> Thank you for your insightful comments and feedback.
>
> **Related work**
>
> We included Thakoor et al (2021) in the related work section of our updated manuscript, as well as added a more in-depth discussion of how previous and concurrent works of graph transformers relate to our model.
>
> **Dataset size**
>
> We agree that more data would most likely lead to performance improvements as well as more stable results. The datasets used were chosen based on three criteria: We used datasets that (1) are publicly available, (2) have been used by previous work to make our model comparable to theirs and (3) come with some form of expert annotation to enable quantitative evaluations. We added this to our limitation section in the updated manuscript.
>
> **Balancing the dataset**
>
> We have not investigated balancing the dataset. Neuronal morphologies are very unbalanced by nature, since cell types are unevenly distributed. Any useful unsupervised learning method should be able to work in such settings. However, we agree that assessing the effects of the unbalance could be an interesting future research direction.
>
> **Equivariant architectures**
>
> Thank you for the idea! Equivariant architectures are indeed an interesting way forward, especially for rotation equivariance. Translation invariance is less of a concern for us, as the neuronal skeletons are normalized to be centered on the origin. We merely use minor translation around the origin as a data augmentation to prevent shortcut learning due to too similar node locations between the two views.
> We added a section regarding equivariant architectures to our limitation section.
>
> **Node feature masking**
>
> In the neuronal skeletons, the only available node features are the xyz-positions of the nodes. The positions of neighboring nodes are highly correlated. Thus, when masking them they can be reconstructed fairly well by simply averaging the positions of the neighboring nodes. Therefore, we believe that node feature masking does not give a strong learning signal in our setting.

---

> > ### Comment · Reviewer_wWM4 · 2023-06-07
> >
> > Thanks for discussing my questions. On node feature masking, a way to circumvent the trivial feature reconstruction by neighbour averaging could simply be to have a bigger mask covering a whole neighbourhood?
> >
> > Anyway, I am happy with the revision and submitted my final recommendation accordingly

---

### Review · Reviewer_Tde6 · 2023-05-22

**Summary Of Contributions:**

The paper proposes a self-supervised representation learning approach (based on DINO [Caron et al., 2021]), that obtains graph-level embeddings using a transformer-based architecture. Using graph data augmentation, e.g., with node subsampling, random rotation, node jittering, etc., the model is trained to maximize similarity between the embeddings for different augmented views of an input. The transformer-based architecture uses a novel, modified "adjacency-conditioned" attention mechanism, which imposes a learned bias on the attention values towards the adjacency matrix of the input graph. Experimental validations are thoroughly performed (e.g., on downstream classification tasks) by learning representations from graph-like neuronal morphologies on five publicly available datasets.

**Audience:**

Yes

**Broader Impact Concerns:**

No specific concern.

**Claims And Evidence:**

Yes

**Requested Changes:**

I think the paper's claims are well supported in the manuscript, and there are no major concerns on my side. Only minor questions below:

- It appears like there is no constraint on the bias factors \lambda and \gamma, which are learned per-node. Would the approach still reasonably work if one reduces it to a single learned parameter vector by setting e.g., \lambda=1-\gamma, and hence reducing the number of parameters in W by D?
- How sensitive is the learning algorithm to the softmax temperature of the teacher encoder?
- Variable p is not defined in the main manuscript or the supplementary until Table C.1 where the chosen values are listed? It is also confusing that it might be referring to the DINO notation in Figure C.4. A clarification on how p relates/differs from z should be added for the reader.
- In Figure 1.B caption, the embeddings are stated to lie on a 32-dim vector space, which is not necessarily fixed and can vary. Since this becomes confusing later, perhaps this should be discussed parameterically in text and clarified in 4.2.2 with the chosen values.
- In Table C.1, the embeddings are chosen to be 64 dimensional for the M1 PatchSeq dataset, although in Section 4.2.2 it was stated that for neuronal datasets it was fixed to 32. This should be corrected. Also since N denotes something else, in Table C.1, N layers -> \# layers, and N heads -> \# heads.
- Typo on page 9, second line of Section 5.5: "1'000" -> "1,000".



**Strengths And Weaknesses:**

Strengths: GraphDINO is generic to any graph-like data structure with similar node features, and appears to not require heavy tailoring for each problem. The transformer attention mechanism is extended feasibly to graph-like structures with a novel and simple adjacency matrix bias. Experimental analyses are performed well in depth and addresses all related aspects of the analyses and proposed model.

Weaknesses: Some minor changes are still necessary for clarification, especially in terms of notation for the architecture.

---

> ### Author Response · Authors · 2023-05-31
> **Authors’ response to Reviewer Tde6**
>
> Thank you for your constructive feedback and acknowledging that our work is novel and thoroughly evaluated!
>
> **Parameterization of bias factors**
>
> In our current setting, the factors $\lambda$ and $\gamma$ are only constrained to be positive, which is enforced by taking the exponential (equation 1). The softmax of the attention afterwards ensures that the attention weights overall sum up to one. One can think of the two parameters as encoding both the softmax temperature and the relative weighting of adjacency and key-value attention. Setting $\lambda = 1 - \gamma$ fixes the temperature.
> We ran an additional ablation experiment, in which we substitute the exponential in equation 1 with the sigmoid function and set $\lambda = 1 - \gamma$ as suggested. This results in an accuracy of  63.4 ± 0% (Mean±STD over three random seeds) for the BBP dataset compared to 65.8 ± 1% with our previous parametrization. The more constrained parametrization results in a slightly reduced accuracy, suggesting that keeping the softmax temperature as a degree of freedom is helpful .
>
>
> **Softmax temperature of teacher encoder**
>
> A low temperature of the softmax of the teacher encoder is required as part of the mechanisms proposed in DINO [1] to prevent collapse of the representation during training. In the DINO paper [1], it is observed that a temperature lower than 0.06 is required to prevent the collapse (see Appendix D, additional ablations).
> In preliminary experiments, we tested temperatures of 0.04 and 0.06 and did not find a significant difference in performance, so we used 0.06.
>
>
> **Text changes**
>
> We have incorporated the suggested edits for clarification in Sec. 3 and 4.2.2 and adapted Fig. 1 and Appendix C.1. We fixed the typo.
>
>
> [1] Caron et al. (2021): Emerging Properties in Self-Supervised Vision Transformers

---

> > ### Comment · Reviewer_Tde6 · 2023-06-06
> > **thanks for the responses**
> >
> > Thanks to the authors for their responses and comments.
> >
> > I find the performed experiments/revisions sufficient, and the corrections that I have suggested seem to be all incorporated. Overall, the claims of the paper appear to be technically correct and the experiments convincingly support this. I submitted my final recommendation accordingly.

---

### Decision · Action_Editors · 2023-06-12

**Recommendation:** Accept as is

**Comment:**

All reviewers agree that the manuscript contains interesting results, in particular for the field of computational neuroscience. It also tackles an interesting input modality that has not been considered in this context. The technical novelty is limited since similar graph transformers exist. Nevertheless, the novel application scenario and strategies to adapt the method to the considered domain are worth publication.

**Audience:**

Yes. The paper is interesting for the audience working on self-supervised representation learning, in particular from a neuroscience perspective.

**Claims And Evidence:**

The claims made in the submission are supported by convincing evidence. All reviewers note that the paper is technically sound.